# End-to-End Deep Neural Network Architectures for Speed and Steering Wheel Angle Prediction in Autonomous Driving

Pedro J. Navarro [1,*], Leanne Miller [1], Francisca Rosique [1], Carlos Fernández-Isla [1] and Alberto Gila-Navarro [2]

1   División de Sistemas e Ingeniería Electrónica (DSIE), Campus Muralla del Mar, s/n, Universidad Politécnica de Cartagena, 30202 Cartagena, Spain; leanne.miller@upct.es (L.M.); paqui.rosique@upct.es (F.R.); carlos.fernandez@upct.es (C.F.-I.)
2   Genética Molecular, Instituto de Biotecnología Vegetal, Edificio I+D+I, Plaza del Hospital s/n, Universidad Politécnica de Cartagena, 30202 Cartagena, Spain; alberto.gilan@um.es
*   Correspondence: pedroj.navarro@upct.es; Tel.: +34-968-32-6546

**Abstract:** The complex decision-making systems used for autonomous vehicles or advanced driver-assistance systems (ADAS) are being replaced by end-to-end (e2e) architectures based on deep-neural-networks (DNN). DNNs can learn complex driving actions from datasets containing thousands of images and data obtained from the vehicle perception system. This work presents the classification, design and implementation of six e2e architectures capable of generating the driving actions of speed and steering wheel angle directly on the vehicle control elements. The work details the design stages and optimization process of the convolutional networks to develop six e2e architectures. In the metric analysis the architectures have been tested with different data sources from the vehicle, such as images, XYZ accelerations and XYZ angular speeds. The best results were obtained with a mixed data e2e architecture that used front images from the vehicle and angular speeds to predict the speed and steering wheel angle with a mean error of 1.06%. An exhaustive optimization process of the convolutional blocks has demonstrated that it is possible to design lightweight e2e architectures with high performance more suitable for the final implementation in autonomous driving.

**Keywords:** autonomous driving; end-to-end architecture; speed and steering wheel angle prediction; DNN for regression

## 1. Introduction

Autonomous driving technology has advanced greatly in recent years, but it is still an ongoing challenge. Traditionally, intelligent decision making systems on board autonomous vehicles have been characterized by their enormous complexity [1] and are composed of multiple subsystems, including a perception system, global and local navigation systems, a control system, a surroundings interpretation system, etc., [2]. These subsystems are combined aiming to cover the complicated decisions and tasks which the vehicle must perform whilst driving. To obtain the objectives of the vehicle, these subsystems use a wide range of techniques which include: cognitive systems [3], agent systems [4], fuzzy systems [5], neural networks [6], evolutionary algorithms [7] or rule-based methods [8].

Deep learning techniques are becoming increasingly popular and are now a valuable tool in a wide range of industries, including the automotive industry, due to their powerful image feature extraction. These techniques have allowed the so-called end-to-end (e2e) driving approach to appear, simplifying the traditional subsystems greatly and reducing the tasks of modeling and control of the vehicle [9] (Figure 1). The appearance of DNNs mean that decision-making systems on board autonomous vehicles can replace many of the subsystems mentioned previously with neural blocks [10]. These neural blocks, properly interconnected and trained with the correct data are capable of obtaining performances greater than 95% for the prediction of vehicle control variables [11]. An advantage of these models is that they generally require fewer onboard sensors as the main source of

information fed to the DNNs usually consists of RGB images and kinematic data from an inertial measurement unit (IMU) [12]. This makes end-to-end driving systems more easily accessible than the traditional perception subsystems with sensors such as LIDAR which are very costly.

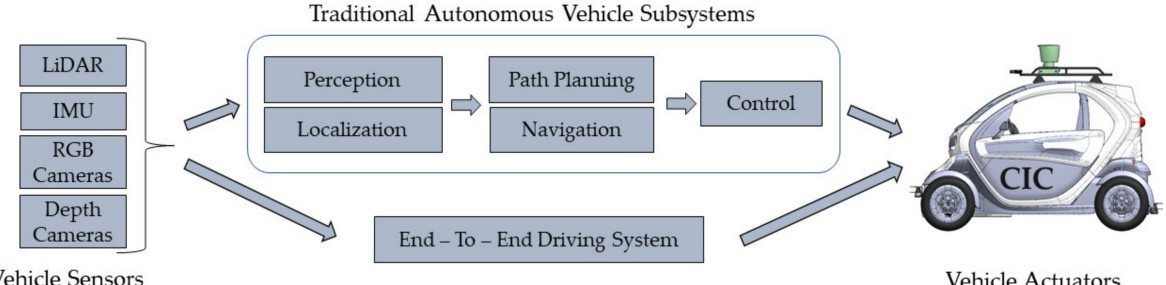

**Figure 1.** Traditional driving systems compared to end-to-end driving system.

Deep learning methods for autonomous driving have gained popularity with advancements in hardware, such as GPUs, and more readily available datasets, both for end-to-end driving techniques [13] and the use of deep learning in individual subsystems [14]. There have been a variety of different approaches for the development of driving applications using end to end learning techniques. In one study, a 98% accuracy was obtained using convolutional neural networks (CNN) to generate steering angles from images generated by a front view camera [15]. In a similar work, a sequence of images from a public dataset was used as input to the CNN, to predict whether the vehicle was accelerating, decelerating or maintaining speed as well as calculating the steering angle [16].

An interesting approach designed a CNN to develop a human-like autonomous driving system which aims to imitate human behavior meaning the vehicle can better adapt to real road conditions [13]. The authors used 3D LIDAR data as input to the model and generated steering and speed commands, and in a driving simulation managed to decrease accidents with the autonomous system ten-fold compared with the human driver. A driving simulator was also used to test a CNN-based closed loop feedback to control the steering angle of the vehicle [17]. The authors designed their own CNN, DAVE-2SKY, using the Caffe deep learning framework and tested the system in a lane-keeping simulation. The results were promising, although problems occurred if the distance to the vehicle in front became less than 9 m.

Various long short-term memory (LSTM) models have also been studied. A convolutional LSTM model with back propagation was trained to obtain the steering angle from video frames using the Udacity dataset [18]. An FCN-LSTM architecture was used to predict driving actions and motion from images obtaining almost 85% accuracy. A convolutional LSTM model was also used to predict steering angles from a stream of images from a front facing camera [19], improving on the results from previous works [20].

Another approach consists in adding more sensors. In one work a dataset was obtained using surround view cameras in addition to the typical front view camera [21]. The data obtained by the cameras was used to predict the speed and steering angle using existing pretrained CNN models. The use of surround view cameras improved the results obtained at low speeds (<20 km/h), but at greater speeds the improvement was less significant.

In this work, we present a detailed study implementing six end-to-end DNN architectures for the prediction of the vehicle speed and the steering wheel angle. The architectures have been trained and tested using 78,011 images from real driving scenarios, which were captured by the Cloud Incubator Car (CIC) autonomous vehicle [2].

## 2. Materials and Methods

DNN end-to-end architectures require large volumes of data for the models to converge correctly. The data needed to create DNN models for autonomous driving or ADAS can be obtained from three different types of sources:

1. Ad hoc tests. To perform this type of testing, large resources are required, in the form of one or more vehicles, expensive perception systems (e.g., LIDAR) and personnel capable of the installation, integration and commissioning of sophisticated sensors and data recording systems. In addition, the data must be post-processed, and the synchronization of the different vehicle information sources is required.
2. Public datasets. There are datasets developed by businesses and universities for autonomous driving where data obtained from the perception systems of their vehicles can be accessed [10]. Some of these present diverse scenarios with different light and meteorological conditions [22].Table 1 shows some recent public datasets including number of samples, types of images available and types of vehicle control actions stored.
3. Simulators. Given the complexity of conducting real tests, autonomous driving simulators have become one of the most widely used alternatives. The simulation industry ranges from simulation platforms, vehicle dynamics simulation and sensor simulation to scenario simulation and even scenario libraries. At present, there are many options, including generic solutions which make use of games and physic engines for simulation [23] and robotics simulators [16]. Recently on the market companies that develop simulation products specifically designed to satisfy the needs of autonomous driving have appeared. Some of these companies include Cognata, CARLA, METAMOTO, etc.

**Table 1.** Public datasets for autonomous driving.

| Ref./Year | Samples | Image Type | LIDAR | RADAR | IMU | Control Actions |
|---|---|---|---|---|---|---|
| UPCT 2019 | 78,000 | RGB, Depth | Yes | No | Yes | Steering wheel, Speed |
| Lyft L5 [24]/2019 | 323,000 | RGB | Yes | No | Yes | - |
| nuScenes [25]/2019 | 1,400,000 | RGB | Yes | Yes | Yes | - |
| Pandaset [22]/2019 | 48,000 | RGB | Yes | No | Yes | - |
| Waymo [23]/2019 | 1,000,000 | RGB | Yes | No | Yes | - |
| Udacity [16]/2016 | 34,000 | RGB | Yes | No | Yes | Steering wheel |
| GAC [26]/2019 | 3,240,000 | RGB | No | No | No | Steering wheel, Speed |

In this work ad hoc data has been chosen. To obtain the data, a custom dataset was created, as the result of ad hoc driving tests performed using the Cloud Incubator Car autonomous vehicle (CICar) [2] (see Figure 2), an autonomous vehicle prototype based on the adaption of the commercial electric vehicle, Renault Twizy. The vehicle has been conveniently modified and houses a complete perception system consisting of a 2D LIDAR, 3D HD LIDAR, ToF cameras, as well as a localization system which contains a real-time kinetic unit (RTK) and inertial measurement unit (IMU, see Figure 2c) and automation of the driving elements of the vehicle (accelerator, brake, steering and gearbox). All of this is complemented with the biometric data of the drivers taken during the driving tests.

### 2.1. Driving Tests

A group of 30 drivers of different age and gender were selected to perform the driving tests, of which five were discarded due to synchronization problems, recording failure or incomplete data. The driving tests were carried out in Cartagena in the Region of Murcia, Spain, following a previously selected route with real traffic.

This route provides a significant set of typical urban driving scenarios: (a) junctions with right of way and changes of priority; (b) incorporation, internal circulation and exiting of a roundabout; (c) driving along a road with and parking areas; (d) merging traffic

situations. In order to contemplate a greater variety of environmental conditions, each driver completed the route twice at different times of day (morning, afternoon or evening). In Figure 3 a sample of some of the dataset images is shown, where some of the different driving conditions captured during the tests can be observed.

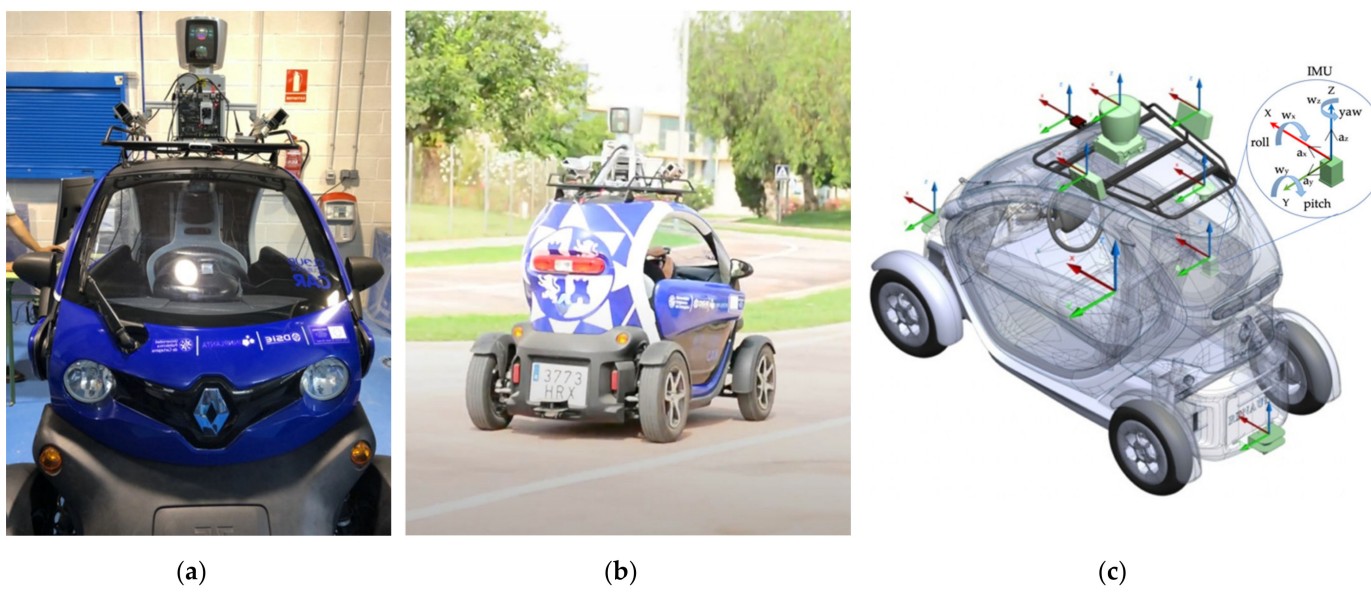

(**a**)　　　　　　　　　　　(**b**)　　　　　　　　　　　(**c**)

**Figure 2.** (**a**) Cloud Incubator Car autonomous vehicle (CiCar). (**b**) CiCar on data acquisition mission. (**c**) Vehicle model and IMU sensor measurement details.

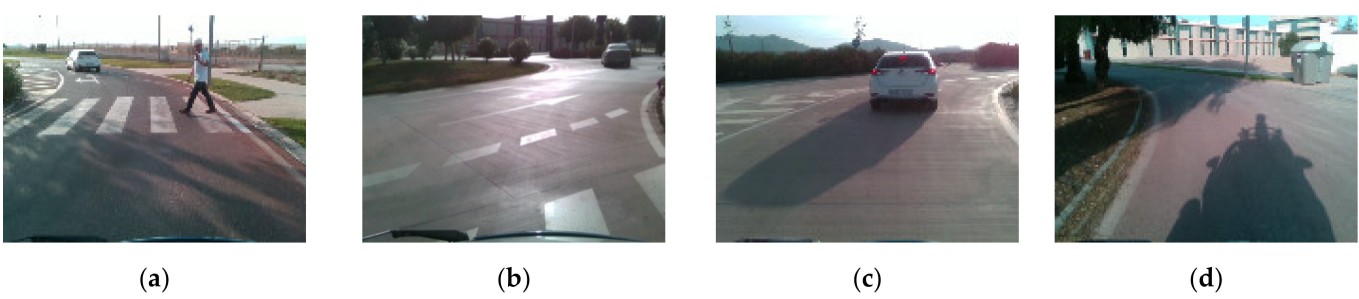

(**a**)　　　　　　　(**b**)　　　　　　　(**c**)　　　　　　　(**d**)

**Figure 3.** Images from dataset. (**a**) pedestrian crossing; (**b**) saturation of the illumination on roundabout; (**c**) car braking; (**d**) complex shadows on the road.

### 2.2. Vehicle Configuration

As mentioned previously, the data was collected using the CICar prototype vehicle in manual mode, driven by a human driver. In Table 2 the variables and data acquired during the driving tests are shown, as well as the information about the devices and systems used to obtain the data.

Each sensor works with its own sample rate, and in most cases this is different between devices. To achieve the correct data synchronization and reconstruct the temporal sequence with precision, stamping times have been generated for each sensor and these have been synchronized at the start and end of the recording. Therefore, all the devices are controlled by the control unit onboard the vehicle, providing a perfect temporal and spatial synchronization of the data obtained by the different sensors. The data from each test is downloaded and stored in the central server once the drive has finished.

**Table 2.** CICar. Sensor data.

| Variable/Unit | Device/System | Frequency |
|---|---|---|
| Vehicle position/(LLA) [1]<br>acceleration/(m/s$^2$)<br>angular speed/($^\circ$/s) | GNSS-IMU | 4 Hz<br>20 Hz<br>20 Hz |
| Steering wheel angle/($^\circ$)<br>Distance/(m)<br>Speed/(m/s) | compactRio Control unit | 50 Hz<br>50 Hz<br>50 Hz |
| Frontal image/<br>Driver attention image<br>Surroundings Cloud Points | RGBD Camera<br>RGB Camera<br>LIDARs, ToF cameras | 25 fps<br>25 fps<br>10 Hz |

[1] LLA—latitude, longitude, and altitude.

### 2.3. Deep Learning End-to-End Architectures Classification

End-to-end (e2e) systems based on DNN architectures applied to autonomous driving can model the complex relationships extracted from the information obtained from the vehicle perception system. This is achieved using different types of neural blocks grouped into layers (e.g., convolutional layers, fully-connected layers, recurrent layers, etc.), with the aim of generating direct control actions on the steering wheel, the accelerator and the brake. These actions on the vehicle control elements can be categorical, e.g., increase or decrease the speed, or they can generate a setpoint on the controller, e.g., turn 13.6 degrees or reach 45 km/h.

The machine learning algorithms that are used to model driving actions belong to the set known as supervised learning. These algorithms acquire knowledge from a dataset of samples previously acquired during driving tests with a previously conditioned vehicle [2] or from driving simulators [27]. These datasets include data from the perception system, such as: images (RGB o IR), LIDAR, RADAR, IMU, as well as the actions performed by the driver on the vehicle control elements, such as the steering wheel, the accelerator and the brake.

The generation of discrete variables by a machine learning algorithm is known as regression and is a widely studied problem [28]. Regression models for DNN use the gradient descent function to search for the optimal weights that minimize the loss function. The loss functions used for these models differ from those used in the classification models, with the most used being the mean absolute error, mean square absolute error or mean absolute percentage error, among others.

This work proposes a classification of e2e architectures based on the type of data received by the DNN from the vehicle perception system. This is done by considering the image provided by the visual perception system of the vehicle as the main data source for the e2e architecture. Based on the type of network input, the architectures have been classified into three types: (1) single data e2e architecture (SiD-e2e), (2) mixed data e2e architecture (MiD-e2e) and sequential data e2e architecture (SeD-e2e).

### 2.3.1. SiD-e2e Architecture

This type of architecture uses a single data source for the input layer to generate the setpoints directly for the control elements of the vehicle. The SiD architectures use the visual information provided by one or more cameras located on the front and periphery of the vehicle to compose a single image of the vehicles field of view of the vehicle as a visual input to the network [15,29,30]. Before being processed by the DNN, the images are reduced in size and normalized. Subsequently, the images go through convolutional layers of different kernel size (k $\times$ k) and depth (d) which allow the image features that minimize the cost function to be extracted automatically in successive layers. After the convolutional layers, the resulting vector is transformed into one dimension (F layer) and connected to a set of fully-connected layers (FC) which have the decision-making capacity. Lastly, the FC layers end in the number of neurons equal to the number of variables to be

predicted [15,28]. Figure 4 shows an example of the SiD architecture where the normalized image feeds a group of convolutional layers with different kernel sizes, followed by a set of fully-connected layers and a final output layer.

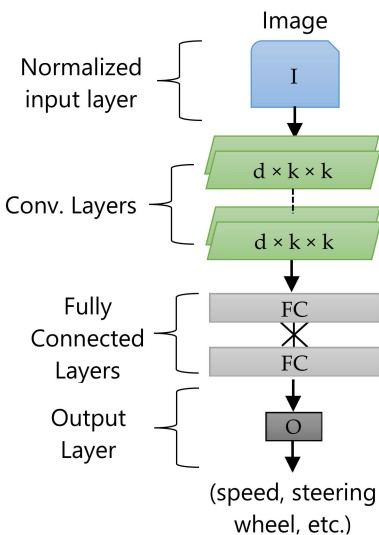

**Figure 4.** Single data e2e architecture (SiD).

The number of convolutional layers, their size, padding and stride, as well as the number of neurons in the FC layers are adjusted empirically. These parameters are dependent on the training dataset and the size of the input images. There are works where the architectures have been designed using banks of convolutional filters of increasing size [30] and there are others where the design is the opposite [31,32]. Generally speaking, the convolutional layers with a small kernel size extract reduced spatial characteristics, such as traffic signs, traffic lights or lane separation lines, while those with a greater kernel size detect larger elements in the image, such as vehicles, pedestrians or the road [31].

### 2.3.2. MiD-e2e Architecture

Mixed data architectures allow different data sources from the vehicle, such as RADAR, longitudinal and lateral accelerations, angular velocities, maps or GPS to be merged together with the visual information from the vehicle's cameras. The inclusion of more information sources in the DNN aims to: (1) improve the performance of the model, (2) improve the prediction of specific cases or abnormal driving; and (3) increase the tolerance to failures produced by the data sources [21,29,33]. As shown in Figure 5, this type of architecture combines the results of the SiD-e2e, such as those shown in the previous Section 2.3.1, with a set of FC layers which allows the mapping of the characteristics from other vehicle data sources on a layer that concatenates all the information.

Figure 5 shows a first input branch where the relevant information is extracted from the image with a second branch that extracts extra information, for example from the IMU or GPS. The concatenation layer receives a specified number of inputs from both branches of the model. The number of connections from each branch is usually determined using empirical techniques. MiD architecture is habitually used in data fusion in the perception systems of autonomous vehicles or ADAS.

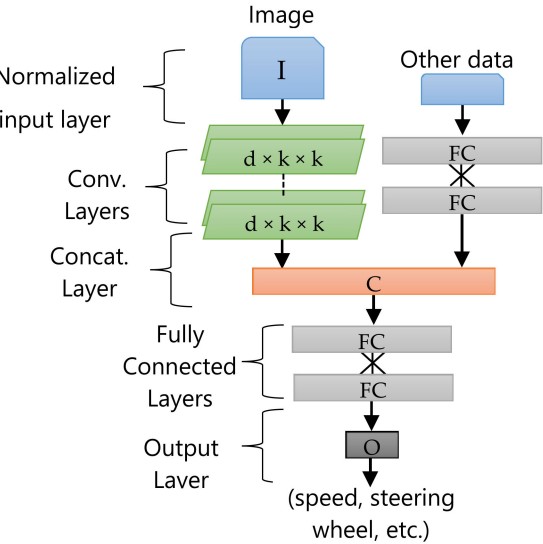

**Figure 5.** Mixed data e2e architecture (MiD).

### 2.3.3. SeD-e2e Architecture

Driving is a task where the future actions on the vehicle's control elements depend greatly on the previous actions, therefore the prediction of the control actions can be modeled as a time series analysis [16,26,34]. Sequential data based architectures aim to model the temporal relationships of the data using feedback neural units (see Figure 6), these types of neural networks are known as recurrent neural networks (RNN) [34]. Basic RNNs can learn the short-term dependencies of the data but they have problems with capturing the long-term dependencies due to vanishing gradient problems [35]. To solve the vanishing gradient problems, more sophisticated RNN architectures have appeared which use activation functions based on gating units. The gating unit has the capacity of conditionally deciding what information is remembered, forgotten or for passing through the unit. The long short-term memory (LSTM) [36] and GRU (gated recurrent unit) are two examples of these kinds of RNN architectures [37].

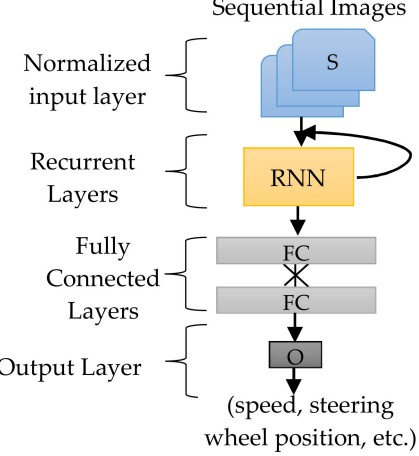

**Figure 6.** Sequential data e2e architecture (SeD).

RNN [15], LSTM (long short-term memory) [16] and GRU (gated recurrent unit) are the most used for modeling the temporal relationships in the field of e2e architectures. The use of RNN in e2e architectures requires the network input data to be transformed into temporal sequences in the form of time steps (*ts*). The partitioning of the *N* input samples

of the network will generate (*N-ts*) temporal sequences that will correspond to an output vector from the network according to Equation (1):

$$S.\ input = \{[I_1, .., I_{ts}], [I_2, .., I_{ts+1}] \ldots, [I_{n-1-ts}, .., I_{N-1}]\},$$
$$output = \{ \qquad o_{ts+1}, \qquad o_{ts+2}, \ldots \ldots, \qquad o_N\} \qquad (1)$$

Figure 7 shows the procedures to generate *N-ts* sequences of size *ts* from a dataset composed of N images and N pairs of output values (v: speed, θ: steering wheel angle).

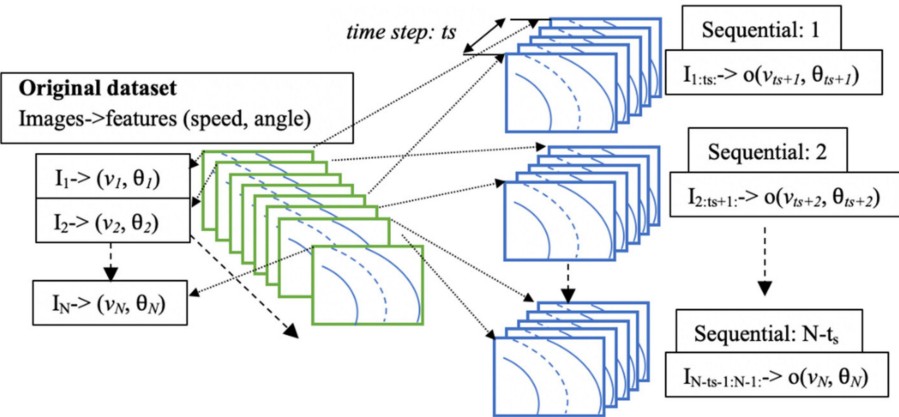

**Figure 7.** Composition of sequential images and output values data.

To create a model from the SeD-e2e architectures, this will be trained with temporal sequences of size *ts* (I$_1$ to I$_{ts}$) and the next output vector to predict (v$_{ts+1}$, θ$_{ts+1}$) as it is shown in the Figure 7.

### 2.4. Paremeters of Deep Neural Network Architectures

The number of parameters which come into play during the design process of a DNN is enormous and we can separate them into three types:

(1) Network input parameters. These parameters refer to the way the network input values are presented. For data in the form of images, the shape parameters include:

- Normalization. Normalization must be performed on the data before training the DNN. An adequate normalization can improve the convergence and performance of the network. Equations (2) and (3) show the most common techniques.

$$Scaled(0,1) = (x_i - min)/(max - min) \qquad (2)$$

$$Standarized(\mu = 0, \sigma = 1) = (x_i - \mu)/(\sigma) \qquad (3)$$

where *min* and *max*, are the maximum and minimum values present in the dataset $X = \{x_1, \ldots, x_N\}$, with *u* and σ being the average and standard deviation of the dataset, respectively. There are other normalization techniques, for example, the mean can be substituted for the mode in Equation (3), for cases in which the data distribution does not align below the mean.
- Resizing. As a general rule and especially in e2e architectures for autonomous driving, the image size is reduced before being processed by the network. The main reason for this is to decrease the network processing time and the resources involved in the prediction.
- Color space transformations. It is common to transform the input image to a color space other than the one supplied by the camera to improve performance, for example HSI, LAB, etc., [10].
- Preprocessing. When the data is captured from different sources or dataset, these tend to have disparate features from the devise itself or from the lighting of

the scene where the images were captured, therefore histogram equalization or image enhancement algorithms are usually applied to normalize the appearance of the entire dataset.
- Data augmentation. This technique consists in increasing the size of the original dataset in order to achieve higher levels of generalization and to improve the performance of the network [38].

(2) Architecture configuration parameters. These parameters constitute the composition of one architecture or another, and these include:
- Type of layer. The architectures can stack different sets of layers in each branch: FC, Convolutional, RNN, Concatenated, etc.
- Layer settings. Each layer has specific configurations, for example, convolutional layers can be configured with different types of filters, $3 \times 3, 5 \times 5, \ldots, k \times k$, their depth or number of layers.
- Layer distribution. The architectures can consist of a single branch, can be branched at the input, or at the output or both.

(3) Hyperparameters. These parameters are characteristic of the learning process implicit in DNN, and among them are:
- Batch size. Number of samples that will be passed through the network before updating the weights.
- Number of Epochs. Number of times the learning algorithm will work with the training dataset.
- Optimization Algorithm. The most widely used is the gradient descent algorithm. However, variations have been developed, such as AdaGrad which adjusts the learning rate. Another variant is Adam, with adaptive momentum.
- Learning rate. The optimization algorithm allows how much the weights are updated at the end of each batch to be controlled.
- Momentum. Controls the modification applied in the instant before updating the weights and influences the weight updates for the current instant.
- Activation function. These functions control the nonlinearity of the neurons individually. The most commonly used activation functions are: softmax, softsign, rectified linear unit, tanh, sigmoid, hard sigmoid, linear, etc., [39].
- Neurons in the hidden layer. This parameter indicates the capacity of the network.

### 2.5. End-to-End Architecture Design

From a top-down design point of view, the process of developing an end-to-end architecture is similar to creating a puzzle made up of different blocks, where each block consists of a set of layers and configuration parameters as explained in Section 2.4. One of the biggest handicaps in the design of DNN architectures appears in the connectivity between the blocks, and this must be solved adequately at the data level for all of them to fit correctly.

In the works reviewed in the literature for the classification of SiD, MiD and SeD architectures (see Sections 2.3.1–2.3.3), groups of layers representing higher level blocks or units can be identified. In these groups, input and output blocks, feature extraction blocks (2D or 3D convolutions), decision-making blocks (fully-connected), concatenation blocks and recurring blocks can be distinguished.

For the design of the e2e architectures developed in this work, seven blocks of different types of layers and parameters with different functionality have been defined as shown in Figure 8.

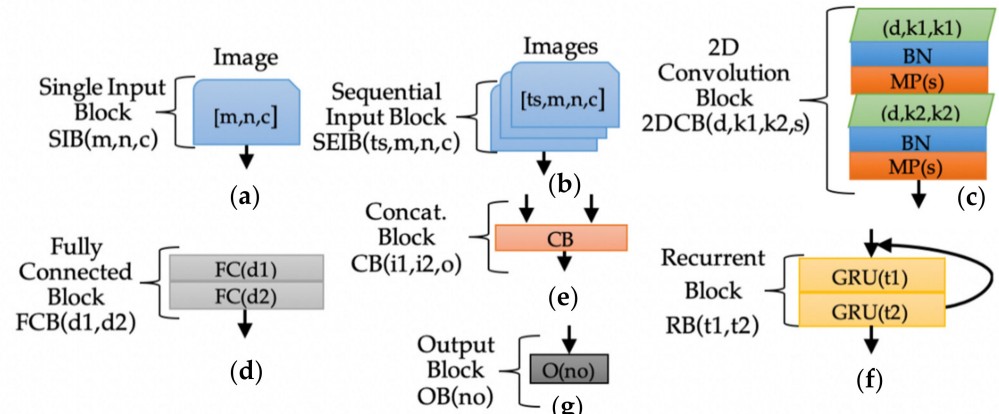

**Figure 8.** High abstraction block for design of e2e architectures: (**a**) single input block (SIB); (**b**) sequential input block (SEIB); (**c**) 2D convolution block (2DCB); (**d**) fully connected block (FCB); (**e**) concatenation block (CB); (**f**) recurrent block (RB); (**g**) output block (OB).

The designed blocks are described below:

(a) Single input block: SIB(m,n,c). This block is made up of a multidimensional input vector that is composed of an image of size [m, n, c] (where m, n and c correspond to rows, columns and channels, respectively) or a one-dimensional vector of data from the vehicle sensors [1, nf], being nf the data size. In both cases, the data must be normalized using Equation (1) or Equation (2).

(b) Sequential input block: SEIB(nt,m,n,c). This block consists of a multidimensional vector containing image sequences of size ts, [ts, m, n, c], as indicated in Figure 7.

(c) 2D Convolutional Block 2DCB(d,k1,k2,s). The 2D convolutional block is dedicated to feature extraction and is composed of eight sequentially arranged layers as follows: F1[Conv2D(d,k1,k1) + Conv2D(d,k1,k1)] + BatchNorm() + MaxPooling2D(s) + F2[Conv2D(d,k2,k2) + Conv2D(d,k2,k2)] + BatchNorm() + MaxPooling2D(s). In the design of this block, two filters F1 and F2 have been inserted, each with 2D convolutional layers (Conv2D). In each filter F1 and F2, (d,k1,k2) corresponds to the kernel depth and size of both 2D filters. Lastly, after each filter, a Batch Normalization layer has been inserted to reduce overfitting and a MaxPooling2D layer to downsample the data, where s is the mask value of the MaxPooling2D layer. The aim of the two double filter sets F1 and F2 is to capture image features of different sizes.

(d) Fully-connected block FCB(d1,d2). This block is composed of two neuron layers which are fully connected sequentially of sizes d1 and d2: FC(d1) + FC(d2)

(e) Concatenation block CB(i,o). The aim of this block is to merge the input i-flatten layers into output o-flatten layers in the MiD architectures.

(f) Recurrent block RB(t1,t2). This type of block is made up of two GRU recurrent layers sequentially connected: GRU(t1) + GRU(t2), t1 and t2 correspond to the number of neurons in each recurrent layer. These units are intended to capture temporal information from the SEIB block image sequences. GRU recurrent layers have been used in this work, but they could be substituted for LSTM or RNN.

(g) Output block OB(no). The output block binds non-output neurons which coincide with the number of values to be predicted by the architecture.

Once the blocks that form the architectures have been defined, the design process has been divided into three stages: 1) Neural block distribution design, 2) Definition of constant and variable parameters and 3) Block optimization with variable parameters.

In the first design stage, and to perform a detailed study of the behavior of the SiD, MiD and SeD architectures for predicting the speed and steering wheel angle (v, θ) for autonomous vehicles, six architectures of two types have been designed which are called: SiD1, SiD2, MiD1, MiD2, SeD1 y SeD2. The difference between type 1 (Figure 9a,c,e) and type 2 (Figure 9b,d,f) lies in the distribution of the output blocks, either in a single branch

or in two independent branches. In the design, the blocks described above have been combined according to the different input and output distributions, as shown in Figure 9.

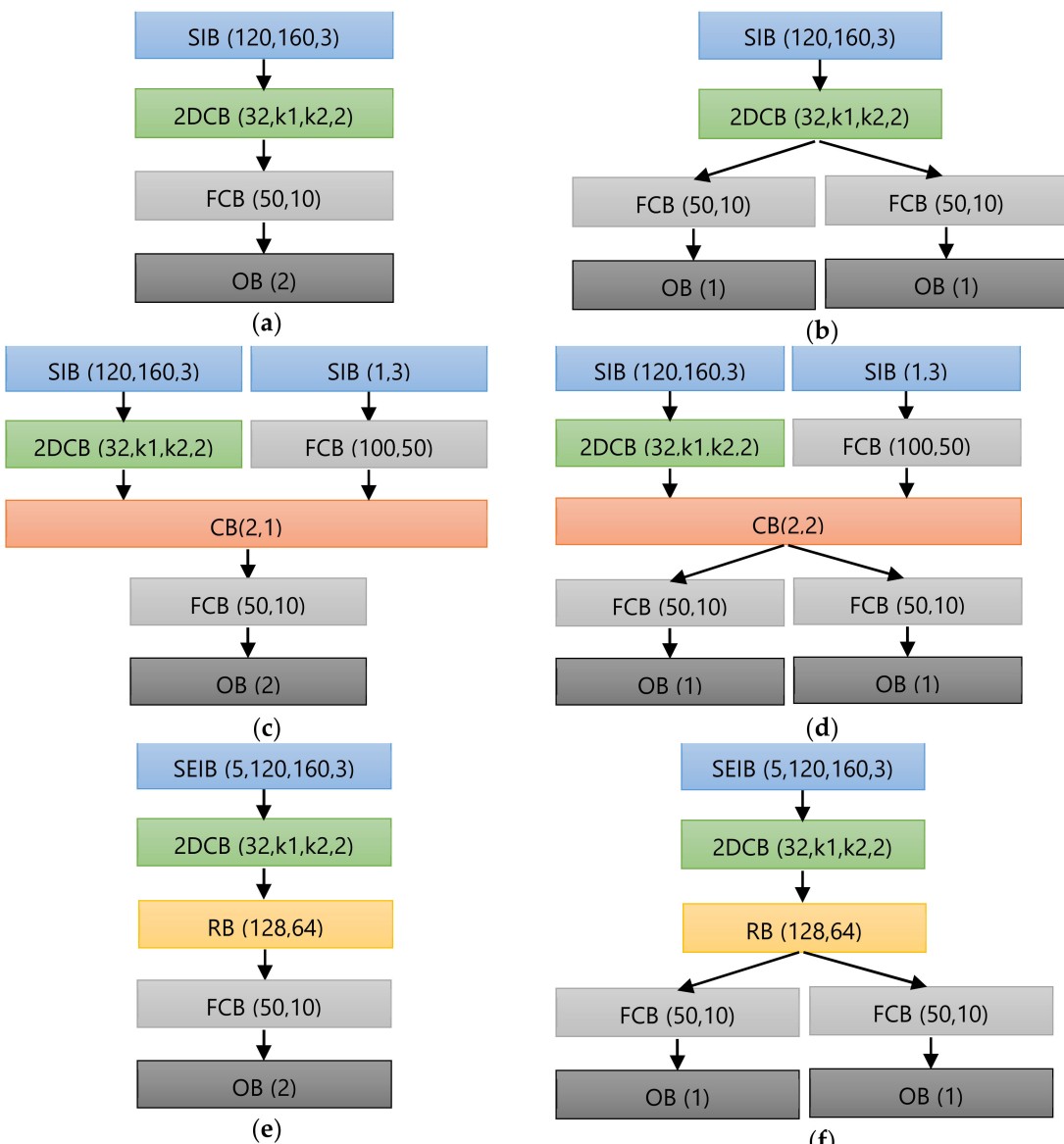

**Figure 9.** Type 1 and type 2 e2e architectures. (**a**) SiD1; (**b**) SiD2; (**c**) MiD1; (**d**) MiD2; (**e**) SeD1; (**f**) SeD2.

In the second design stage, common values have been established for all blocks of the six architectures, with the aim of obtaining lightweight architectures, that is, minimizing the number of parameters. The configurations chosen for each architecture are as follows:

1.  The SiD1 and SiD2 architectures have been configured with the same parameters which are SIB(120,160,3), 2DCB(32,k1,k2,2) and FCB(50,10). The different output layer distribution causes the difference in results from SiD1 with OB(2) (Figure 9a) and SiD2 with OB(1)+ OB(1) (Figure 9b).
2.  The MiD1 and MiD2 architectures share blocks with the SiD model above: SIB (120,160,3), 2DCB(32,k1,k2,2), SIB(1,3) and FCB(50,100). However, after the concatenation block, MiD1 has a branch consisting of [FCB (50,10) + OB(2)] (Figure 9c) and in MiD2 it branches into two parts [FCB (50,10) + OB(1)] (Figure 9d)
3.  The SeD1 and SeD2 architectures have been configured in a similar manner but with a sequential input SEID(5,120,160,3), 2DCB(32,k1,k2,2), FCB (50,10) and RB(128,64).

In the output layers, SeD1 consists of [FCB (50,10) + OB(2)] (Figure 9e) and MiD2 separates into two branches [FCB (50,10) + OB(1)] (Figure 9f).

In the final design stage and with the objective of obtaining an optimal parameter configuration for the six architectures, two tests have been designed:

(1)   Optimization of 2DCB(32,k1,k2,2). In this test the six architectures were evaluated with different combinations of k1 and k2:

   a.   k1 $\leq$ k2: k1 with a constant value of 3 and k2 variable between 3 and 29 in increments of 2.

   b.   k1 > k2: k1 with a constant value of 29 and k2 variable between 3 and 15 in increments of 2.

The test calculates which combination of k1 and k2 sizes are able to extract the best spatial features from the input images. The variation of the sizes of k1 and k2 will determine if the performance of the models increases when initially using small or large kernels.

(2)   (Optimization of the MiD1 and MiD2 models. This test is used to determine the influence of the vehicle dynamics parameters in the prediction of the models. This is performed using the discrete acceleration and angular velocity values obtained by the IMU on the XYZ axes of the vehicle, together with the input images.

## 3. Results

The implementation of the models has been performed using the Keras 2.4.1 libraries under TensorFlow 2.4.0. The six architectures proposed were programed in Python 3.6 with the Spyder 4.1.5 IDE running on a computer installed with Windows 10 Professional with Intel Core™ processor i9-10800K at 4.20 GHz, 64 GB DDR4 and three dedicated graphics NVIDIA GeForce RTX 2070, RTX 3070 (Santa Clara, CA, USA) and RTX3090 (Taipei, Taiwan).

### 3.1. Model Configuration

To train the models, a dataset of 78,011 samples has been used (see Section 2.1), consisting of discrete values of speed in km/h, steering wheel angle (degrees), XYZ accelerations (m/s$^2$), XYZ angular velocities ($°$/s) together with color images of size 160 × 180 × 3. The discrete values have been normalized using Equation (3) and the images using Equation (2). The dataset has been shuffled and has been divided into two groups:

1.   Training dataset, which consists of 58,508 samples, 75% of the total samples.
2.   Validation dataset, which consists of 19,503 samples, 25% of the total samples.

Table 3 shows the hyperparameters used to configure the training and validating phases of the models.

**Table 3.** Hyperparameters configuration.

| Parameter Name | Value |
| --- | --- |
| Batch size | 10 |
| Number of Epochs | 50 |
| Optimization algorithm | Adam |
| Loss function | Mean Square Root Error |
| Metric | Mean Absolute Error |
| Learning rate | 0.001 |
| Momentum | None |
| Activation function in 2DCB | Relu |
| Activation function in OB | Linear |

### 3.2. Performance Metrics

To evaluate the performance of the proposed architectures, the metrics mean absolute error were calculated during the training and validation of the models. The mean absolute

percentage error (MAPE see Equation (4)) was used to compare the speed and steering wheel angle obtained by the models. All data supplied in this section were calculated using the validation dataset.

$$\text{MAPE} = 100 \times MAE/spanV \tag{4}$$

Being *MAE*, the mean absolute error of the prediction (speed or angle) and *spanV* the difference between the maximum value and the minimum value of variable to predict (or span). The span of the speed and steering wheel angle are 58.05 km/h and 841°, respectively.

- Convolutional block Optimization for k1 ≤ k2

Table 4 shows the values obtained when calculating the MAPE metric for the speed and steering wheel angle predictions during the optimization process of the convolutional blocks for k1 ≤ k2 for all the proposed architectures.

**Table 4.** MAPE obtained during the 2DCB optimization k1 ≤ k2 for the architectures. Bold texts show the best results.

| k1 ≤ k2 | | SiD1 | SiD2 | MiD1 | | MiD2 | | SeD1 | SeD2 |
|---|---|---|---|---|---|---|---|---|---|
| 2DCB(d,k1,k2,s) | - | - | - | axyz | wxyz | axyz | wxyz | - | - |
| 2DCB(32,3,3,2) | v | 13.13% | 67.21% | 4.69% | 49.32% | 2.58% | 12.59% | 10.93% | 16.71% |
| | θ | 2.09% | 13.20% | 1.43% | 0.87% | 0.78% | 3.46% | 1.78% | 2.81% |
| 2DCB(32,3,5,2) | v | 10.22% | 67.24% | 3.85% | 49.33% | 2.36% | 12.57% | 18.76% | 12.15% |
| | θ | 1.66% | 13.23% | 1.19% | 0.87% | 0.74% | 3.41% | 3.10% | 1.90% |
| 2DCB(32,3,7,2) | v | 9.56% | 6.01% | 3.29% | 49.32% | 2.23% | 2.12% | 18.03% | 14.58% |
| | θ | 1.49% | 13.21% | 0.96% | 0.87% | 0.64% | 0.48% | 3.13% | 2.36% |
| 2DCB(32,3,9,2) | v | 9.25% | 67.20% | 3.45% | 49.31% | 12.53% | 1.71% | 13.93% | 15.24% |
| | θ | 1.42% | 13.26% | 0.99% | 0.87% | 3.41% | 3.41% | 2.20% | 2.55% |
| 2DCB(32,3,11,2) | v | 7.82% | 67.30% | 3.04% | 49.33% | 12.52% | 1.66% | 15.52% | 15.27% |
| | θ | 1.18% | 13.29% | 0.87% | 0.87% | 1.03% | 0.49% | 2.50% | 2.31% |
| 2DCB(32,3,13,2) | v | 7.24% | 67.21% | 2.74% | 32.14% | 2.30% | 1.91% | 12.86% | 16.46% |
| | θ | 1.37% | 0.94% | 0.79% | 0.71% | 0.72% | 0.51% | 1.86% | 2.59% |
| 2DCB(32,3,15,2) | v | 6.54% | 67.23% | 3.00% | 33.97% | 2.32% | 1.99% | 16.74% | 10.19% |
| | θ | 1.04% | 1.56% | 0.85% | 0.69% | 0.70% | 0.55% | 2.63% | 1.37% |
| 2DCB(32,3,17,2) | v | 6.52% | 6.58% | 2.46% | 49.36% | 2.50% | 1.82% | 9.12% | 10.64% |
| | θ | 1.10% | 0.79% | 0.76% | 0.87% | 0.69% | 0.43% | 1.30% | 1.62% |
| 2DCB(32,3,19,2) | v | 7.01% | 5.24% | 2.55% | 49.33% | **1.94%** | 1.96% | 7.48% | 7.16% |
| | θ | 1.17% | 13.32% | 0.73% | 0.87% | **0.65%** | 0.52% | 0.97% | 1.01% |
| 2DCB(32,3,21,2) | v | 5.72% | 67.06% | 2.91% | **6.94%** | 12.51% | 1.76% | 10.15% | 8.30% |
| | θ | 0.80% | 0.88% | 0.87% | **0.13%** | 0.75% | 0.44% | 1.41% | 1.06% |
| 2DCB(32,3,23,2) | v | 5.75% | 67.05% | **2.19%** | 7.24% | 2.54% | 1.72% | 5.67% | 5.42% |
| | θ | 0.86% | 0.70% | **0.69%** | 0.15% | 0.70% | 0.46% | 0.76% | 0.67% |
| 2DCB(32,3,25,2) | v | **5.36%** | 4.82% | 2.23% | 9.08% | 2.11% | **1.69%** | 5.52% | 6.12% |
| | θ | **0.84%** | 0.59% | 0.68% | 0.14% | 0.64% | **0.43%** | 0.64% | 0.72% |
| 2DCB(32,3,27,2) | v | 5.65% | 4.77% | 2.30% | 8.16% | 2.63% | 1.79% | 4.58% | 4.88% |
| | θ | 0.82% | 0.55% | 0.71% | 0.16% | 0.77% | 0.44% | 0.60% | 0.56% |
| 2DCB(32,3,29,2) | v | 6.19% | **4.19%** | 2.46% | 7.23% | 2.09% | 1.85% | **4.29%** | **2.66%** |
| | θ | 0.89% | **0.45%** | 0.74% | 0.14% | 0.62% | 0.46% | **0.51%** | **0.35%** |
| Minimal errors per variable | v | 5.36% | 4.19% | 2.19% | 6.94% | 1.94% | **1.66%** | 4.29% | 2.66% |
| | θ | 0.80% | 0.45% | 0.68% | **0.13%** | 0.62% | 0.43% | 0.51% | 0.35% |
| Minimal Error per 2DCB | (v, θ) | 3.10% (32,3,25,2) | 2.32% (32,3,29,2) | 1.44% (32,3,23,2) | 3.54% (32,3,21,2) | 1.30% (32,3,19,2) | **1.06% (32,3,25,2)** | 2.40% (32,3,29,2) | 1.51% (32,3,29,2) |

Generally, as can be observed in Table 4, all architectures obtain a lower percentage error for the steering wheel angle prediction than for the speed prediction, which is logical since it is simpler for the models to relate the geometric features from the images (lines, obstacles, traffic signs or the road itself) with the steering wheel rotation angle than with the speed of the vehicle. The architecture with the lowest percentage error for speed prediction is MiD2_wxyz with 1.66% (MAE: 0.96 km/h), with 2DCB(32,3,21,2). The lowest percentage error for the steering wheel angle prediction is MiD1_wxyz with 1.06% (MAE: 1.09°), with 2DCB(32,3,25,2).

The architecture with the lowest mean percentage error for the joint prediction of speed and steering wheel angle with 1.06% (MAE: 0.96 km/h, 3.61°) is MiD2_wxyx, with 2DCB(32,3,25,2).

Regarding the type of architecture (type 1: SiD1, MiD1, SeD1 or type 2: SiD2, MiD2, SeD2), the last row of Table 4 shows that the type 2 architecture design with two independent output branches always obtains a lower combined error in prediction.

Figure 10 shows the trend for the error committed by the models in the prediction of speed for the type 1 (SiD1, MiD1, SeD1 see Figure 10a) and type 2 (SiD2, MiD2, SeD2 see Figure 10b) architectures, with respect to the kernel size k2 × k2 for both types of architectures. As can be observed in the speed prediction, the error tends to decrease as the size of the kernel k2 × k2 increases, for both types of architecture.

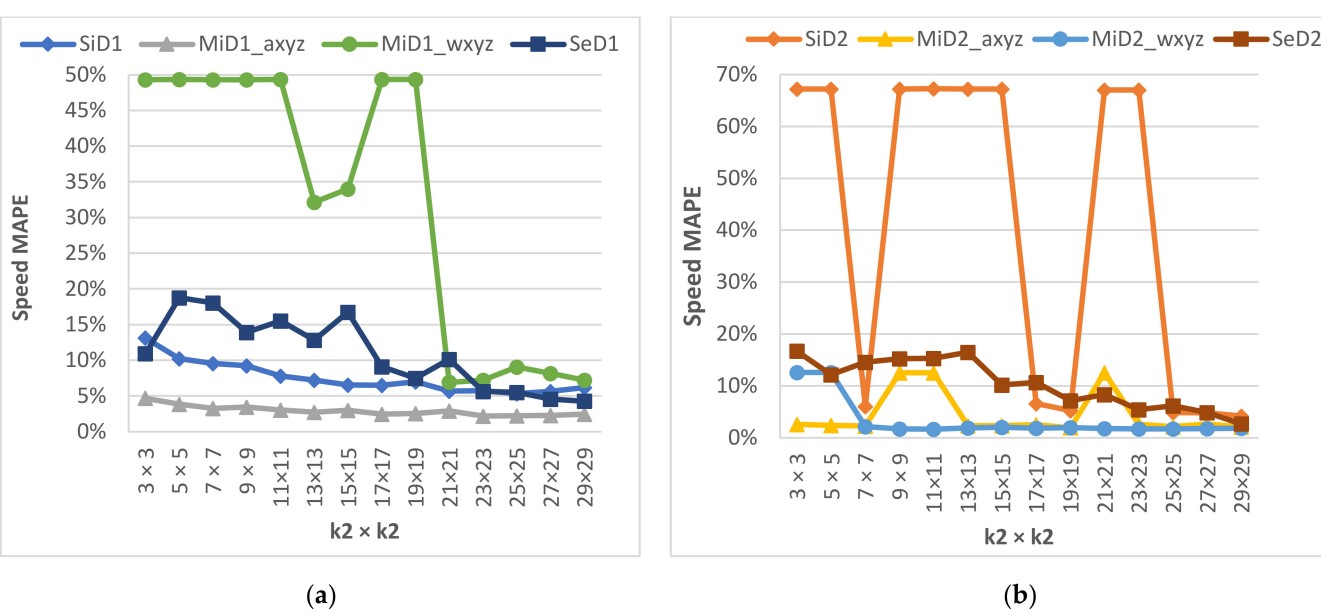

**Figure 10.** MAPE trends for speed prediction of the architectures type 1 and type 2 with convolutional blocks k1 ≤ k2: (**a**) SiD1, MiD1_axyz, MiD1_wxyz, SeD1; (**b**) SiD2, MiD2_axyz, MiD2_wxyz, SeD2.

Figure 11 shows the trend of the errors made by the models for the steering wheel angle prediction for both architectures, type 1 (SiD1, MiD1, SeD1 see Figure 11a) and type 2 (SiD2, MiD2, SeD2 see Figure 11b) according to the kernel size k2 × k2. As was the case for the speed, the error tends to decrease as the kernel value k2 increases.

- Convolutional block Optimization for k1 > k2

Table 5 shows the values obtained using the MAPE metric for the speed and steering wheel angle predictions during the convolutional block optimization process for k1 > k2 in the proposed architectures. As can be observed, the lowest error for the speed prediction is 2.21% (MAE: 1.28 km/h) and was achieved with the MiD1_wxyz architecture with 2DCB(32,29,3,2). However, the lowest error for the steering wheel angle prediction was given by the MiD2_wxyz architecture with 2DCB(32,29,13,2).

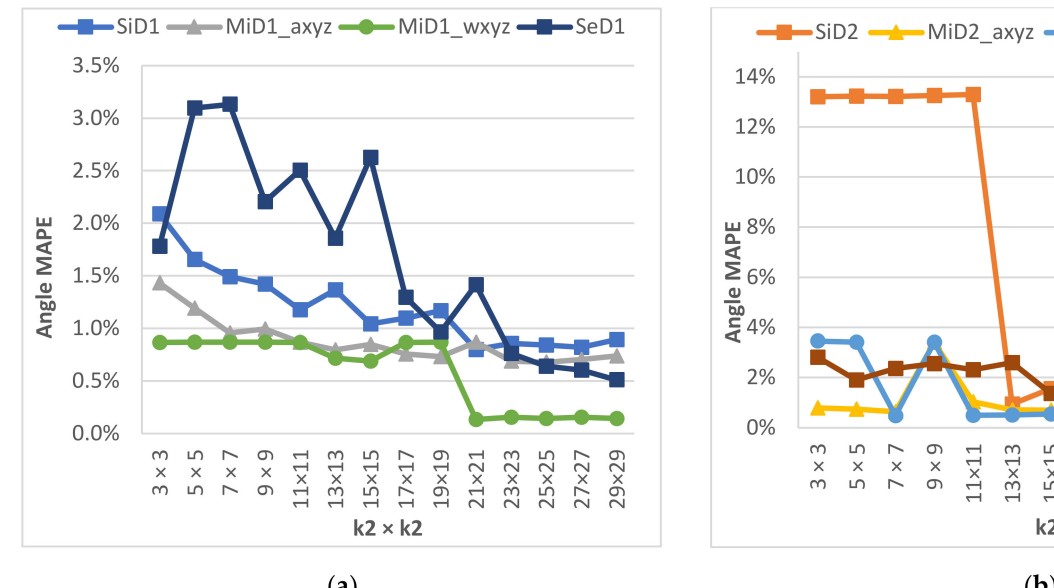
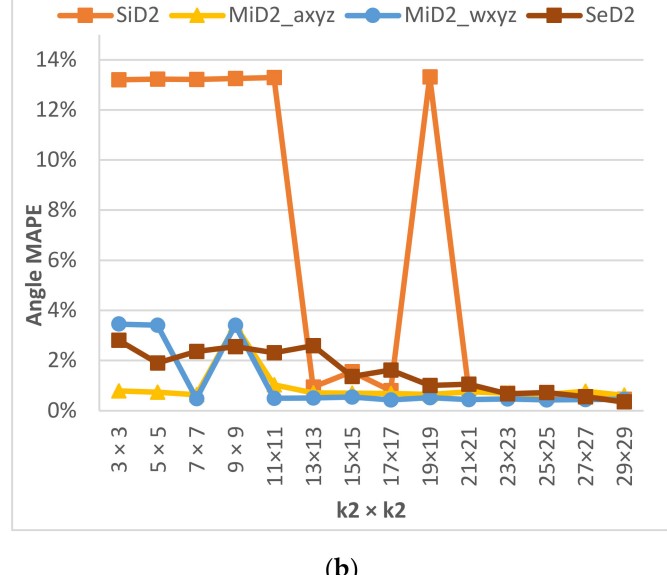

      (**a**)                                     (**b**)

**Figure 11.** MAPE trends for steering wheel angle prediction of the architectures type 1 and type 2 with convolutional blocks k1 ≤ k2: (**a**) SiD1, MiD1_axyz, MiD1_wxyz, SeD1; (**b**) SiD2, MiD2_axyz, MiD2_wxyz, SeD2.

**Table 5.** MAPE obtained during the 2DCB optimization k1 > k2 for the architectures. Bold texts show the best results.

| k1>k2 | - | SiD1 | SiD2 | MiD1 | | MiD2 | | SeD1 | SeD2 |
|---|---|---|---|---|---|---|---|---|---|
| **2DCB(d,k1,k2,s)** | **-** | **-** | **-** | **axyz** | **wxyz** | **axyz** | **wxyz** | **-** | **-** |
| 2DCB(32,29,3,2) | v | 94.09% | 15.41% | 3.45% | 49.36% | 2.81% | **2.21%** | **7.68%** | 15.41% |
| | θ | 8.19% | 2.53% | 1.02% | 0.87% | 0.80% | **0.80%** | **1.13%** | 2.53% |
| 2DCB(32,29,5,2) | v | 11.76% | 6.84% | 3.38% | 15.19% | 2.82% | 2.63% | 8.04% | 9.51% |
| | θ | 1.89% | 0.81% | 1.02% | 0.30% | 0.93% | 0.93% | 1.18% | 1.47% |
| 2DCB(32,29,7,2) | v | 12.38% | 6.81% | **3.11%** | 22.66% | **2.47%** | 2.56% | 13.21% | 11.07% |
| | θ | 1.92% | 0.74% | **0.95%** | 0.81% | **0.78%** | 0.78% | 1.93% | 1.46% |
| 2DCB(32,29,9,2) | v | 9.68% | 7.31% | 3.18% | 11.76% | 2.58% | 2.42% | 9.46% | 7.77% |
| | θ | 1.75% | 1.08% | 0.93% | 0.28% | 0.78% | 0.78% | 1.42% | 0.99% |
| 2DCB(32,29,11,2) | v | 9.77% | **6.65%** | 3.39% | 9.77% | 2.67% | 2.67% | 8.09% | 9.31% |
| | θ | 1.49% | **0.83%** | 0.94% | 0.21% | 0.80% | 0.80% | 0.99% | 1.29% |
| 2DCB(32,29,13,2) | v | **8.96%** | 7.00% | 3.33% | **9.36%** | 3.06% | 2.85% | 8.09% | 7,51% |
| | θ | **1.51%** | 0.77% | 1.01% | **0.19%** | 0.81% | 0.81% | 0.99% | 0.89% |
| 2DCB(32,29,15,2) | v | 10.04% | 7.96% | 3.24% | 9.47% | 3.39% | 2.73% | 9.78% | **7.47%** |
| | θ | 1.66% | 0.89% | 0.93% | 0.21% | 0.89% | 0.89% | 1.38% | **0.91%** |
| Minimal errors per variable | v | 8.96% | 6.65% | 3.11% | 9.36% | 2.47% | **2.21%** | 7.68% | 7.47% |
| | θ | 1.49% | 0.74% | 0.93% | 0.19% | 0.78% | 0.78% | 0.99% | 0.89% |
| Minimal Error per 2DCB | (v, θ) | 5.24% (32,29,13,2) | 3.74% (32,29,11,2) | 2.03% (32,29,7,2) | 4.78% (32,29,13,2) | 1.63% (32, 29,7,2) | **1.51%** **(32,29,3,2)** | 4.41% (32,29,3,2) | 4.19% (32,29,15,2) |

    The lowest mean percentage error for the speed and steering wheel angle predictions combined is 1.51% (MAE: 1.28 km/h, 6.74°) obtained by the MiD2_wxyx architecture with 2DCB(32,29,3,2).

    Figure 12 shows the trend of the errors obtained by the models for the speed prediction for the type 1 and type 2 architectures for k1 > k2. Figure 12a shows that the type 1 architectures have a constant error (<10%) with kernel sizes equal to or greater than 9 × 9. Furthermore, it can be clearly observed that the MiD1_axyz architecture obtains the lowest error for the speed prediction. Regarding the type 2 architectures (SiD2, MiD2, SeD2),

Figure 12b shows that the type 2 architectures have similar behavior to those of type 1, except for the SeD2 architecture which has a decreasing error as k2 × k2 increases.

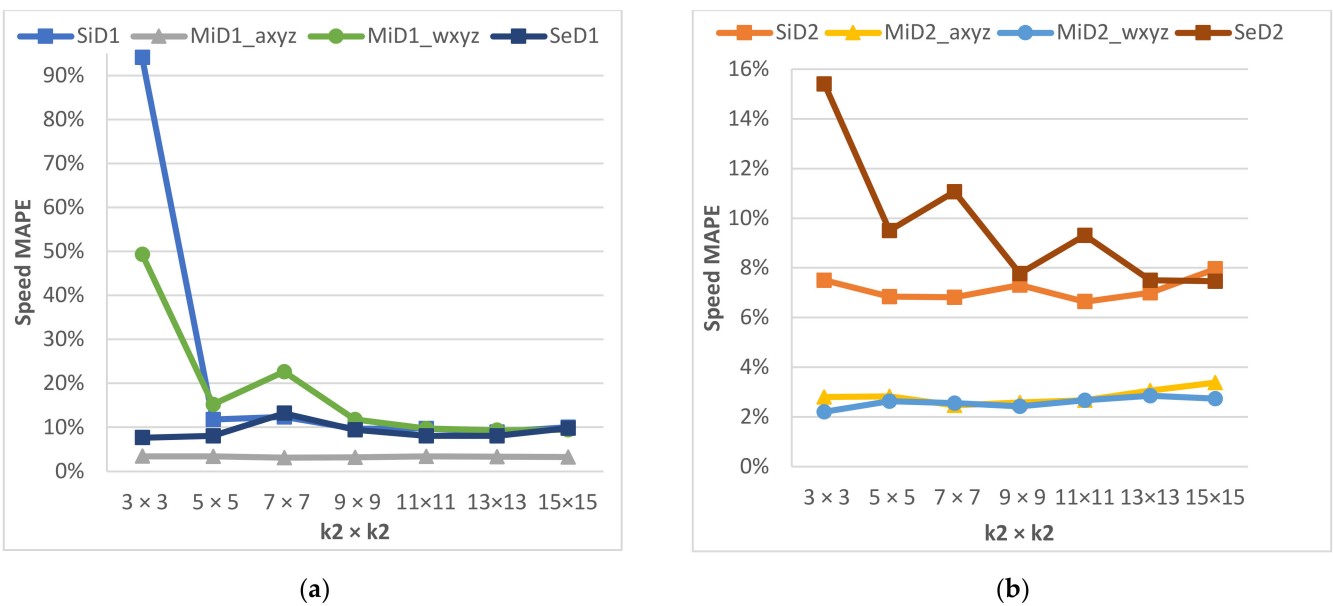

(**a**)

(**b**)

**Figure 12.** MAPE trends for speed prediction of the architectures type 1 and type 2 with convolutional blocks k1 > k2: (**a**) SiD1, MiD1_axyz, MiD1_wxyz, SeD1; (**b**) SiD2, MiD2_axyz, MiD2_wxyz, SeD2.

Figure 13 shows the trend of the MAPE in percentage made by the models in the steering wheel angle prediction for type 1 and type 2 architectures. In Figure 13a,b it can be verified that all architectures maintain the prediction error (<2%) if the kernel size is greater or equal to 5 × 5. The architectures that obtain the lowest error are MiD1_wxyz and MiD2_wxyz.

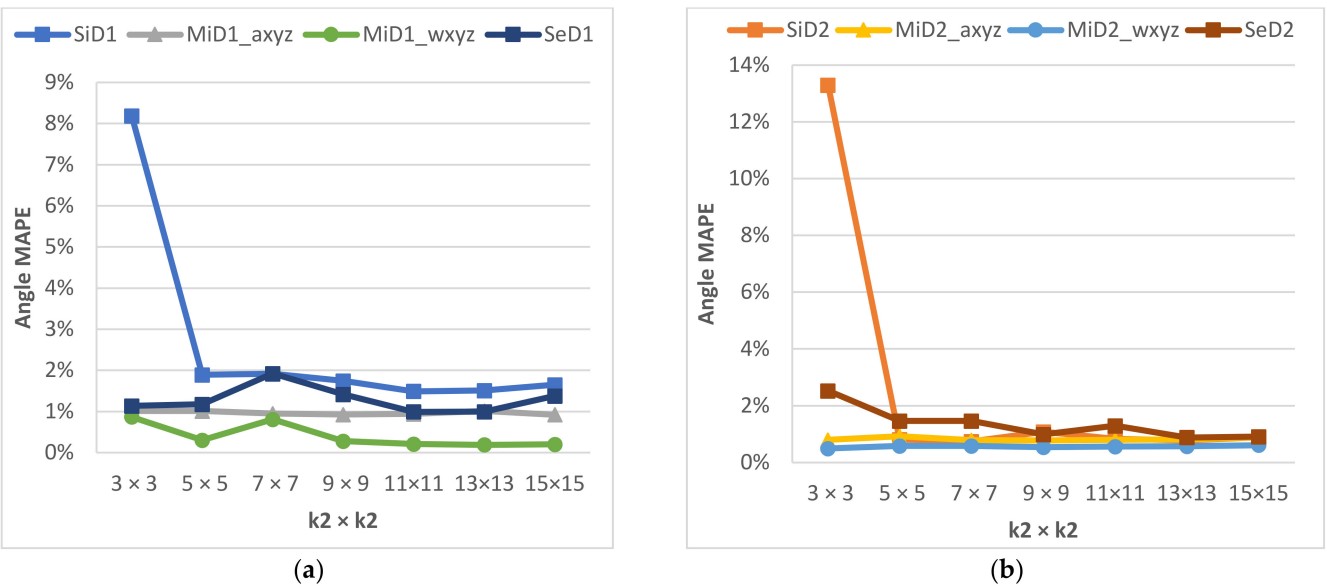

(**a**)

(**b**)

**Figure 13.** MAPE trends for steering wheel angle prediction of the architectures type 1 and type 2 with convolutional blocks k1 > k2: (**a**) SiD1, MiD1_axyz, MiD1_wxyz, SeD1; (**b**) SiD2, MiD2_axyz, MiD2_wxyz, SeD2.

### 3.3. Complexity of the Architectures

The number of parameters for each architecture is an indicator of its complexity. Figure 14 shows the average number of parameters obtained for the SiD, MiD and SeD architectures in the optimization tests for kernel sizes k1 ≤ k2 (Figure 14a) and k1 > k2 (Figure 14b). The use of kernels k2 > k1 drastically reduces the number of parameters for the architectures as can be observed in Figure 14b compared to Figure 14a.

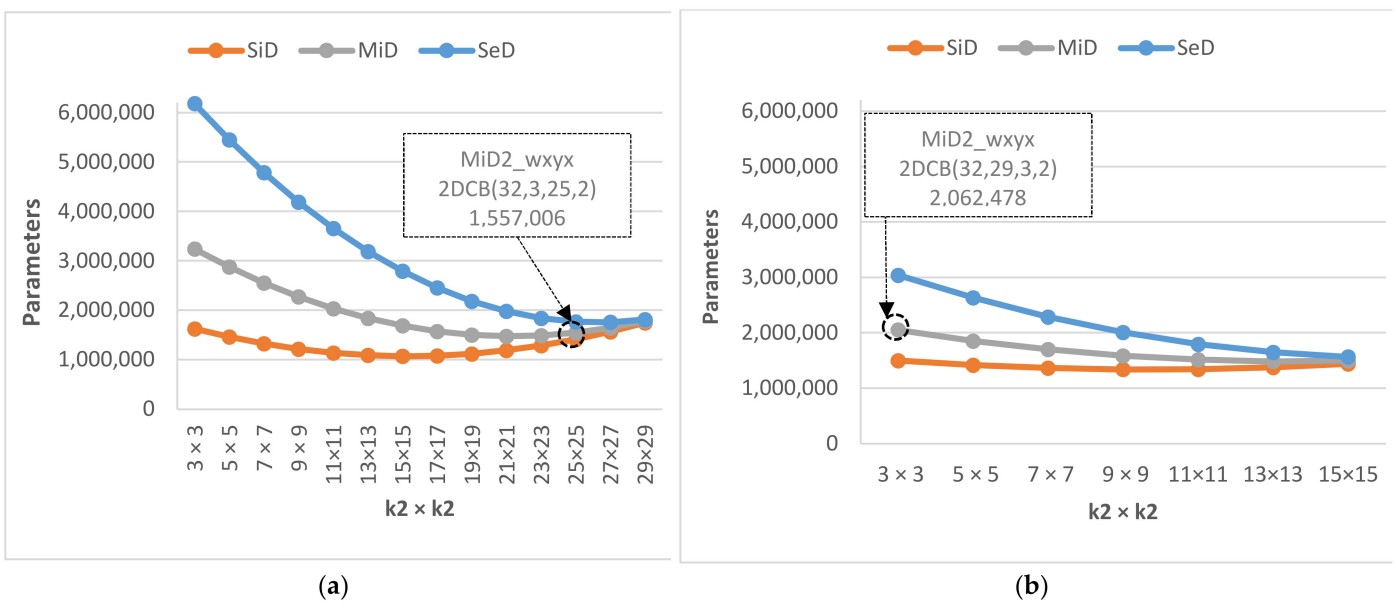

(**a**) (**b**)

**Figure 14.** Number of parameters for architectures SiD, MiD y SeD: (**a**) size kernel k1 ≤ k2; (**b**) size kernel k1 > k2.

As shown in Table 4, the architecture that offers the lowest error in the optimization test for k1 ≤ k2 was MiD2_wxyx with 2DCB(32,3,25,2) and 1,557,006 parameters were used (Figure 14a). In the k1 > k2 optimization test, Table 5 shows that the best results were obtained by the MiD2_wxyx architecture with 2DCB(32,29,3,2) and 2,062,478 parameters (Figure 14b). As can be observed, an adequate optimization process produces better results with fewer architecture parameters.

### 4. Discussion

In the literature review of [16,19–21,26,30,40], compared to the results achieved in this work, a lack of uniformity in the metrics were used and a lack of clarity were found when presenting the results. In some cases, subjective ad hoc metrics that are difficult to reproduce were performed, for example, in [31] a metric called "steering autonomy" is used, which quantifies the number of times the driver had to attend to the autonomous system which was presented as the sole results of the work. In other studies, the set of samples on which the metrics have been calculated are not specifically indicated, nor the number of samples used for training, validation and test sets [15,16,30]. In this work, and to compare the architectures designed, the works which use the mean absolute error (MAE) metric have been selected for the comparison.

Table 6 shows a set of models used for the prediction of control actions orientated at autonomous driving based on end-to-end architectures. In [15,30], one of the pioneering works in the development of end-to-end architectures in the field of autonomous driving is shown. The model was called PilotNet and was tested in [16] with the Udacity dataset for the prediction of the steering wheel angle, obtaining a MAE of 4.26°. The work performed in [29] uses a PilotNet based architecture with RGB images and depth images as input, managing to reduce the error in the angle prediction to 2.41°. In [16], a SeD architecture is presented consisting of two blocks, CNN and LTSM. The proposed model was tested with images from the Udacity database (20 min of video) converted to the HSV color space

and a database created by the authors (SAIC). The average error obtained in the speed and steering angle prediction was of 1.15 km/h and 0.71°, respectively. The authors in [26] compare the performance of the speed and steering wheel angle prediction performance of three architectures, one of MiD type and two of SeD type based on recurring LSTM networks. The comparison is carried out on two databases: the Guangzhou Automotive Cooperate (GAC) and the autonomous driving simulation platform of Grand Theft Auto V (GATV). The best results were an average error of 2.86 km/h and 2.87° with the third architecture proposed.

**Table 6.** MAPE metric obtained during the 2DCB optimization for the SiD1, SiD2, MiD1, MiD2, SeD1 and Se2 architectures.

| Authors | Architecture | Dataset | Input | Output | MAE Results | Parameters |
|---------|--------------|---------|-------|--------|-------------|------------|
| [31,32] | SiD PilotNet | Udacity | RGB front camera | steering angle | 4.26° | 250,000 |
| [35] | MiD | Ad-hoc dataset | 2 RGB + 1 Depth images | steering angle | 2.41° | 156,059 |
| [26] | SeD | Udacity, SAIC | HSV front camera | (speed, steering angle) | (1.15 km/h, 0.71°) | $>25 \times 10^6$ |
| [27] | SeD | GAC, GTAV | RGB images | (speed, steering angle) | (2.86 km/h, 2.87°) | 10,437,744 |
| Proposed | MiD2_wxyz (32,3,35,2) | 78,011 samples | RGB images + IMU data | (speed, steering angle) | (0.96 km/s, 3.61°) | 1,557,006 |

Finally, of the architectures proposed in this work, the architecture which obtained the lowest average error (1.06%) was MiD2_wxyz with the 2DCB(32,3,25,2) configuration. As can be observed, the use of the IMU information, specifically the angular velocity data in the x, y and z axes, together with the use of a small kernel ($k1 = 3$) followed by a large kernel ($k2 = 25$) notably improved the vehicle speed prediction with a MAE of 0.96 km/s. Regarding the angle prediction, a MAE of 3.61° was obtained, which corresponds to an error of 0.43% in the measurement span.

The last column of Table 6 shows the number of parameters of each one of the architectures studied. As can be seen, an excess of complexity represented by the number of parameters does not imply the best result. The architecture that presents the best result in the steering wheel angle prediction is the one given by [16] and uses more than 25 M parameters. However, the best result obtained for the speed prediction is obtained by the proposed architecture MiD2_wxyz(32,3,35,2), with less than 1.5 M parameters. An optimized design of the convolutional layers, such as those developed in this work, shows that low complexity and high performance can be achieved.

## 5. Conclusions

In this work, a novel classification of end-to-end architectures (SiD, MiD and SeD) capable of generating control actions directly on the maneuvering elements of a vehicle has been presented. In addition, the stages for a modular and detailed design of end-to-end architectures for prediction of the speed and steering wheel angle have been presented. To validate the proposed classification, six architectures have been implemented and evaluated and it has been concluded that:

(1) The type 2 architectures, with an independent output branch for each variable, obtain better results in the optimization tests carried out.

(2) In the design of the convolutional blocks, a better performance was obtained using an initial block size of $k1 \leq k2$. Furthermore, the increase in the size of $k_2$ gave a better performance for high values of around eight times $k_1$.

In the comparison between the proposed architectures, it was observed that the use of vehicle dynamics information from the IMU with RGB images improves the speed and steering wheel angle predictions. The inclusion of the angular velocity substantially improved the speed prediction and a mean prediction error of 1.06% was obtained. In

the comparison with other works (see Table 6), the MiD2_wxyz architecture with the 2DCB(32,3,25,2) configuration obtained a better result for the speed prediction in terms of MAE and a 0.43% of error in the angle prediction over the measurement span.

The optimization process performed on the convolutional layer kernels achieved a lighter architecture with excellent performance results.

Future work will continue to explore new architectures that incorporate new and relevant information, such as depth images and LIDAR or RADAR information to improve performance. In addition, the complexity of the database obtained is intended to be increased with more complex driving scenarios.

**Author Contributions:** Conceptualization, P.J.N., L.M., F.R., C.F.-I. and A.G.-N.; methodology, P.J.N., L.M., F.R., C.F.-I. and A.G.-N.; software, P.J.N. and A.G.-N.; validation, P.J.N., L.M. and A.G.-N.; formal analysis, P.J.N., L.M., F.R., C.F.-I. and A.G.-N.; investigation, P.J.N., L.M., F.R., C.F.-I. and A.G.-N.; resources, P.J.N. and L.M.; data curation, P.J.N., L.M., F.R. and C.F.-I.; writing—original draft preparation, P.J.N., L.M. and F.R.; writing—review and editing, P.J.N., L.M., F.R. and C.F.-I.; visualization, P.J.N., L.M., F.R. and A.G.-N.; supervision, P.J.N. and L.M.; project administration, P.J.N.; funding acquisition, P.J.N. and C.F.-I. All authors have read and agreed to the published version of the manuscript.

**Funding:** This work was partially supported by DGT (ref.SPIP2017-02286) and GenoVision (ref.BFU2017-88300-C2-2-R) Spanish Government projects, and the "Research Programe for Groups of Scientific Excellence in the Region of Murcia" of the Seneca Foundation (Agency for Science and Technology in the Region of Murcia—19895/GERM/15).

**Data Availability Statement:** The dataset is available upon request from corresponding author.

**Acknowledgments:** We would like to thank Antonio Padilla Urrea for his contribution to the data acquisition process during the autonomous vehicle missions.

**Conflicts of Interest:** The authors declare no conflict of interest.

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
