# Peer review of "End-to-End Deep Neural Network Architectures for Speed and Steering Wheel Angle Prediction in Autonomous Driving"

_electronics, doi:10.3390/electronics10111266_

Round 1
Reviewer 1 Report
The subtitle 2.2. Deep learning end-to-end architectures, is actually 2.3.
I recommend correcting the numbering of the subtitle and its associated paragraphs.
The conclusions do not contain a comparative analysis (supported by specific data from the tables and figures) of the proposed architectures and other existing architectures highlighting their novel elements.
The bibliography contains 45 current references, but:
- Two of these ([38] and [40]) are not current (1997, 2001);
- The references ([21], [22] and [23] is not in the text.
I recommend correcting the errors related to the bibliography.
Reviewer 2 Report
- The authors have not provided any information regarding the characteristics of the object they control in autonomous driving. The object is a system that has many features of a dynamic nature that affect the vehicle control process (steering wheel speeds and angles). The observation is confirmed by the erroneous recording of vehicle data, as accelerations and angular velocities are labeled in an identical XYZ manner. This proves the ignorance of the description of the object, which is the object of control. The range of changes of the quantities characterizing the dynamic features of the motor vehicle should be given - at least a reference to some publication giving a mathematical model of the car. The presented considerations apply only to electronic processes and can be used for various objects.
- The obtained results are random quantities and it would be necessary to determine the ranges of changes for a given value of the independent variable (graphs in Figures 10 to 14).
Reviewer 3 Report
The author proposed an architecture based on an end-to-end deep neural network for speed and steering wheel angle prediction in autonomous driving. It deals with an important topic. The level of English is fair. The flow of the text is good. The scientific value is fair.
Strong points
- The author presented a classification of end-to-end architectures (SiD, MiD and SeD) capable of generating control actions directly on the manoeuvring elements of a vehicle. They presented an end-to-end architecture for the prediction of the speed and steering wheel angle. have been presented. They implemented six architectures and their results found promising.
In contrast to the existing approach their approach, the MiD2_wxyz architecture with the 2DCB(32,3,25,2) configuration 572Electronics 2021, 10, x FOR PEER REVIEW 20 of 22 obtained a better result for the speed prediction in terms of MAE and a 0.43% of error in the angle prediction over the measurement span.
Potential Improvements
- I wonder if the datasets of only 78,011 images can generalise the proposed approach. An autonomous driving system needs a huge amount of data in order to work flawlessly. With this very low amount of dataset, you may get a good result. However, despite training with a huge amount of data the real-time self-driving cars facing many challenges in predicting the steering angle more specifically predicting the speed. For future work, my suggestion you to train your model with Lyft L5 [21]/2019’s datasets or nuScenes [22]/2019 as these datasets are the size of average size. After that training with these datasets then compare the result and you may have good experience with working a low and huge amount of data.
Minors
- Section 2.2.4 title : paremeters of deep neural network architectures----> parameters of deep neural network architectures
- Kindly double-check for grammatical mistakes such as commas, sentence structures, and spell mistakes.
Round 2
Reviewer 2 Report
The XYZ system in automotive vehicles is assumed to be a right-angled rectangular coordinate system. If we talk about accelerations or linear velocities, the vectors of these quantities lie on the coordinate axes. If we are talking about accelerations and angular velocities then they are around the axes of the XYZ rectangular system. You need to explain exactly what the manufacturer has provided in effect, then you will know what dynamic characteristics of the motor vehicle will be taken into account.
